# Cyberbullying Involvement and Depression among Elementary School, Middle School, High School, and University Students: The Role of Social Support and Gender

**DOI:** 10.3390/ijerph20042835

**Published:** 2023-02-06

**Authors:** Michelle F. Wright, Sebastian Wachs

**Affiliations:** 1Department of Psychology, DePaul University, Chicago, IL 60604, USA; 2National Anti-Bullying Research and Resource Center, Dublin City University, D09 AW21 Dublin, Ireland; 3Department of Educational Studies, University of Potsdam, 14476 Potsdam, Germany

**Keywords:** cyberbullying, age, gender, depression, social support

## Abstract

One aim of this study was to investigate differences in cyberbullying involvement (i.e., victimization, bystanding, perpetration) across four age groups, including 234 elementary school students (4th and 5th grades; 51% female), 363 middle school students (6th through 8th grades; 53% female), 341 high school students (9th through 12th grade; 51% female), and 371 university students (all four years; 60% female). Another aim was to examine the age group differences in the associations between cyberbullying involvement and depression, as well as the moderating effect of social support from parents and friends. Participants completed questionnaires on cyberbullying involvement, depression, and social support from parents and friends. Findings revealed that middle school students were more often involved in cyberbullying as victims, bystanders, and perpetrators, followed by high school and university students, and elementary school students. High school and university students did not differ on their cyberbullying involvement. Gender moderated these relationships for elementary school students, with boys more often involved in cyberbullying perpetration and victimization than girls. In addition, female university students witnessed cyberbullying more so than males. Social support from parents buffered against the negative effects of cyberbullying involvement on depression across all age groups. Results were similar for social support from friends, but only for middle school and high school students. Gender did not influence the associations among age groups, cyberbullying involvement, and depression. The results have implications for designing prevention and intervention programs and ensuring that such programs consider age.

## 1. Introduction

Information and communication technologies (ICTs) are an integral part of our modern society. ICTs have brought many conveniences into our lives, but ICTs have led to exposure to risks [1,2]. Cyberbullying involvement is one risk associated with ICT use, and it is often linked to depression [3,4,5,6]. Because of the positive relationship between cyberbullying involvement, specifically victimization, perpetration, and witnessing, as well as depression, researchers are concerned with identifying factors that might mitigate the negative impacts of cyberbullying. Some factors might involve age groups and perceived social support from parents and friends. Few investigations have focused on age differences in cyberbullying involvement, including age and gender interactions. In addition, although depression is often linked with cyberbullying involvement among various age groups, it is unclear whether the relationship between depression and cyberbullying involvement might change based on the age group. Therefore, the purpose of the present research was to examine age group differences in cyberbullying involvement, as well as the interaction between age groups and gender differences. The next purpose of the research was to examine age group differences in the moderating effect of perceived social support in the positive relationship between cyberbullying involvement and depression.

### 1.1. Cyberbullying Defined

Defined as witnessing, experiencing as a victim, or perpetrating embarrassing and/or intimidating repetitive and hostile behaviors through ICTs [5,7,8,9], cyberbullying has similar elements to traditional face-to-face bullying [5,9,10]. Cyberbullying often includes an imbalance of power between the bully and the perpetrator; it also includes a technical component. Furthermore, cyberbullying also involves repetition, like traditional face-to-face bullying, although ICTs increase the complexity of cyberbullying behaviors. For example, perpetrators can target a victim or victims by posting a humiliating and embarrassing video online; this content has the possibility of being forwarded, with malintent or without, to an almost endless supply of witnesses, who could also choose to forward the content one more time or an infinite number of times. Many cyberbullying behaviors include rumor spreading, harassment, social exclusion, humiliation, physical threats, gossip, and verbal insults. There are also physical forms of cyberbullying, specifically hacking, as well as making anonymous phone calls, forwarding explicit videos, identity threat or pretending to be someone else, and harassment through instant messenger, social media, and text messages [11,12].

### 1.2. Age

Because of the age differences in face-to-face bullying involvement, another predictor of cyberbullying involvement is age. In this literature, elementary school students are more likely to perpetrate physical forms of aggression, more so than those in middle and high schools [13]. Given the increases in verbal skills and the increasing importance of interacting with peers, verbal and relational forms of aggression become more salient in middle school. Furthermore, adolescents develop better skills at navigating and understanding social situations. Often conceptualized as an indirect form of bullying, cyberbullying involvement could also increase during adolescence and potentially decrease during late adolescence and young adulthood [14].

Expecting cyberbullying involvement to increase during adolescence is further complicated by findings that indicate cyberbullying involvement is often associated with greater ICT use, which might further increase children’s and adolescents’ susceptibility to witnessing cyberbullying [3,10,15]. The age of children when they first use technologies is decreasing, and some researchers have found incidences of cyberbullying as young as nine years old [16]. There are few longitudinal studies on whether age might increase or decrease the risks associated with perpetrating, experiencing, and/or witnessing cyberbullying.

Some research has found that cyberbullying is highest among early adolescents, in comparison to children, late adolescents, and young adults [17]. Furthermore, one study found that hacking forms of cyberbullying increased during the middle school years and later declined during the high school years [18]. Unfortunately, age has been found to be an inconsistent predictor of cyberbullying involvement. Wade and Beran [19] reported that cyberbullying perpetration and victimization were highest among 9th graders in high school and lower among those adolescents in middle school. Many reasons might account for why age has proven to be an inconsistent predictor of cyberbullying. For example, the amount of time spent online, and the types of technologies used could be a better explanation for cyberbullying involvement than age, as many studies do not control for duration of use and types of technologies utilized. As a result of not controlling for use and types of technologies, the researchers might really be studying age-related increases in technology use versus age-related trends. In addition, many studies do not consider the interaction of gender and age.

### 1.3. Gender

Gender has also been considered a predictor of cyberbullying involvement. However, like age, gender has also proven to be an inconsistent predictor of cyberbullying. Research has revealed that boys have more technological skills, in comparison to girls [4]. Technological skills are positively correlated with cyberbullying involvement. Some studies have revealed that boys perpetrate cyberbullying more often than girls [4,5,20,21,22] and that girls more often report victimization by cyberbullying [23,24,25,26,27,28]. Other studies have revealed no gender differences in cyberbullying involvement [29,30,31,32,33]. Research on gender and multiple age group differences in cyberbullying involvement have not been conducted to our knowledge.

### 1.4. Depression

Cyberbullying involvement, including perpetration, victimization, and witnessing, is associated with depression. The negative consequences associated with cyberbullying involvement have increased researchers’, educators’, and parents’ concerns with cyberbullying involvement. Research has revealed that cyberbullying victimization is related negatively to lower global happiness, school happiness, school satisfaction, family satisfaction, and self-satisfaction [34]. Furthermore, cybervictims also report greater feelings of anger, fear, and sadness, when compared to uninvolved children and adolescents [23,29,35]. They also report more internalizing problems (e.g., depression and anxiety), as well as externalizing problems (e.g., violence, drug use/abuse) [3,4,5,6]. Available research also indicates that school problems are more often found among children and adolescents who are involved in cyberbullying. Lower school functioning (e.g., academic performance) is positively correlated with cyberbullying involvement, measured one year later [36].

Critics of the earlier research on cyberbullying involvement and negative consequences argued that these studies did not account for face-to-face bullying involvement, which might have been driving associations found in the literature. To address this gap in the literature, the joint effects of both traditional face-to-face bullying and cyberbullying involvement should be considered. Findings from this literature revealed that victims of both forms of bullying reported more internalizing difficulties, in comparison to adolescents who experienced only one type of victimization [37]. This literature has also grown to include studies that examine the relationship between cyberbullying involvement and psychological, academic, and behavioral consequences while controlling for traditional face-to-face bullying involvement. Consequently, researchers must account for face-to-face bullying involvement as it is highly correlated with cyberbullying involvement. Bonanno and colleagues [38], after controlling for face-to-face bullying perpetration and victimization, found that cyberbullying involvement was associated with depression and suicidal ideation, beyond the impacts of face-to-face bullying involvement.

### 1.5. Perceived Social Support

Social support refers to the knowledge that someone is cared for, respected, and belongs to a network of people who are concerned with one’s welfare [39,40,41]. The knowledge that someone is available to provide physical, social, and psychological support during negative events or situations has powerful impacts, such as increasing feelings of security and self-worth. Parents are often a major source of social support, as well as friends, who become increasingly relied on for social support, as children become adolescents and young adults [42]. Ample research has provided support for the positive impact of social support on the involvement in face-to-face bullying. In this research, bullying victims often have lower quality friendships, which has direct impacts on their help and protection [41,43,44,45]. Victims are often isolated from social interactions, leading to poor social relationships, which increase their vulnerability to bullying. Another study found that low perceived social support from family and peers were positively associated with face-to-face victimization [46].

Similar to the research on face-to-face bullying involvement, perceived social support diminishes the risk of cyberbullying involvement [1,18,34,36,47,48]. In one study, Williams and Guerra [18] found that cyberbullying involvement was lower when adolescents believed they had friends who cared about them. Another study found similar patterns [48]. Furthermore, having a social companion, someone to spend time with, and someone who offers comfort reduced the risk of cyberbullying involvement (Navarro et al., 2013). Family social support protected adolescents from cyberbullying involvement, even when their friends were not supportive [1,47]. Smokowski and colleagues [47] also found that perceived parental support had the strongest reduction in cyberbullying involvement, when compared to support from friends.

Few studies have been conducted on the buffering effects of perceived social support from parents and friends in the associations between cyberbullying involvement and depression. In the literature on face-to-face bullying involvement, research [40] has found that social support buffered against the negative effects of depression for bullying victims. Holt and Espelage [41] found that victims with moderate social support from parents had lower levels of depression, when compared to victims with low perceived social support from parents.

It is currently unknown how age group might influence the buffering effects of perceived social support from parents and friends in the relationships among cyberbullying involvement and depression. Few studies have focused on age differences in social support and how social support mitigates negative outcomes, such as depression, associated with cyberbullying involvement. A study on age differences in social support seeking found that older adults (ages 60+) sought social support less often than young adults (18–25 years;) [49]. Higher perceived support was related to fewer mental health issues among younger veterans (*M* = 37.00, *SD* = 6.00) and older veterans (*M* = 81.70, *SD* = 3.20; [50]. Similar patterns were found in another study on the mediating effects of perceived social support on mental health during the coronavirus [51]. In particular, Cao and colleagues [51] found that the effect of perceived social support on mental health problems was mediated by psychological capital during the pandemic, but only for young adolescents (10–12 years) and not emerging adults (18–25 years). Ultimately, it is unknown whether there are age differences in the buffering effect of social support from friends and parents in the associations between cyberbullying involvement and depression.

### 1.6. The Present Study

The mixed findings regarding the age differences in cyberbullying involvement make it difficult to determine hypotheses. Similarly, it is tough to propose how age and gender might interact to increase or decrease cyberbullying perpetration, victimization, and witnessing due to a lack of research on this topic. Thus, the first purpose of the present study was to investigate age and gender differences in cyberbullying involvement. Following this premise, this research question was proposed:

(1) What are age and gender differences in cyberbullying involvement (i.e., perpetration, involvement, witnessing), while controlling for technology use and face-to-face bullying involvement?

Considering the literature on cyberbullying involvement and social support and age differences in social support, it might be likely that higher social support might reduce the positive relationships between cyberbullying involvement and depression for adolescents in the middle school years than among other age groups. Low social support might be expected to increase the positive relationships among cyberbullying involvement and depression. The second purpose of the present study was to examine the relationships among cyberbullying involvement and depression, as well as the moderating effect of perceived social support from friends and family. The following research questions were developed for the second purpose of the study:

(2) What is the relationship between cyberbullying involvement (i.e., perpetration, victimization, witnessing) and depression, and what are the age differences in these associations, while controlling for technology use and face-to-face bullying involvement?

(3) What, if any, moderating effect does perceived social support from friends and family have on the associations among cyberbullying involvement and depression), and what are the age differences in these moderating effects, while controlling for technology use and face-to-face bullying involvement?

## 2. Materials and Methods

### 2.1. Participants

There were 1309 participants enrolled in this study from middle-class suburbs of a large Midwestern city in the United States. Of these participants, 234 (*M*_age_ = 10.43, *SD*_age_ = 0.10) were elementary school students (51% female) in either the 4th (100 total) or 5th grade (134 total); 363 (*M*_age_ = 13.03, *SD*_age_ = 0.13) were middle school students (53% female) in the 6th (105 total), 7th (115 total), or 8th (143) grades; 341 (*M*_age_ = 16.29, *SD*_age_ = 0.67) were high school students (51% female) in the 9th (86 total), 10th (76 total), 11th (82 total), and 12th (98 total) grades; and 371 (*M*_age_ = 19.98, *SD*_age_ = 0.89) university students (60% female), who were either freshmen (81 total), sophomores (91 total), juniors (86 total), and seniors (108 total). Overall, most participants identified as white (70%), followed by Latinx (20%), Black/African American (5%), Asian (1%), and other (4%). No other income data were collected.

### 2.2. Procedures

Ethical approval was granted before data collection sites were contacted. One elementary school, one middle school, and one high school were used for recruitment from a single school district. Recruitment for the university students involved sharing information about the study through emails sent by instructors of psychology courses and by posting flyers in the psychology building. Incentives were not offered to any of the participants.

Parental permission slips were sent home to elementary, middle, and high school students, unless participants were at least 18 years old, and in that case, informed consent documents were distributed. Of the elementary school parental permission slips, 60 were not returned, 12 were returned without permission, and 234 were returned with permission. For middle school students, 67 parental permission slips were unreturned, 20 did not have permission, and 373 were returned; however, on the day of data collection, 10 middle school students were unavailable (e.g., moved, suspended, in-school suspension) and they were dropped from the study, yielding a total participation rate of 363. For high school students, 10 parental permission slips were returned without permission, 49 were never returned, and 330 parental permissions slips were returned with permission.

Data were collected during the spring of 2019 from all participants. For university students, they completed informed consent prior to completing the questionnaires in a laboratory setting on their school’s campus. Elementary school through high school students participated in school during regular school hours. All participants completed paper and pencil questionnaires on their demographic background, including age (What is your age?), gender (What is your gender?), and technology use, face-to-face bullying and cyberbullying involvement (i.e., perpetration, victimization, bystanding), depression, anxiety, and perceived social support from parents and friends.

### 2.3. Measures

Technology use. Ten items (e.g., How often do you send/receive text messages?) were used to assess participants’ technology use. Items were rated on a scale of 1 (never) to 5 (all the time). Items were combined to form a final score on technology use, with higher indicating greater technology use. Cronbach’s alphas ranged from 0.83 to 0.93.

Face-to-face bullying and cyberbullying involvement. Thirty-six items were included on this questionnaire to ask participants how often they experienced face-to-face bullying and cyberbullying victimization (16 items—8 for each type of victimization; e.g., was insulted online/offline by someone, were called mean names online/offline); face-to-face bullying and cyberbullying perpetration (16 items—8 for each type of perpetration; e.g., insulted someone online/offline, called someone mean names online/offline); and witnessing face-to-face bullying and cyberbullying (16 items—8 for each type of witnessing; witnessed someone being insulted online/offline, witnessed someone being called mean names online/offline) [33]. The items were rated on a scale of 1 (never) to 5 (all the time). All participants answered items according to what they have experienced, perpetrated, and/or witnessed within the current school year. The items for each subscale were averaged separately to form scores for face-to-face and cyber victimization, perpetration, and witnessing. Higher scores indicate greater levels of face-to-face and cyberbullying victimization, perpetration, and witnessing. Cronbach’s alphas ranged from 0.81 through 0.93 for all participants and subscales.

Depression. The Center for Epidemiological Studies - Depression questionnaire was used to assess participants’ depressive symptoms within the past two weeks [52]. Twenty items (e.g., I was bothered by things that usually don’t bother me, I did not feel like eating, my appetite was poor) were included in this questionnaire to measure depression. Items were rated on a scale of 0 (rarely or none of the time) to 3 (most or all of the time). Cronbach’s alphas ranged from 0.80 to 0.86.

Perceived social support from parents and friends. Two subscales were used to access participants’ perceived social support from friends (12 items; e.g., my friends understand my feelings) and parents (12 items; e.g., my parent or parents show they are proud of me; [53]). All items were rated on a scale of 1 (never) to 6 (always). Items for each subscale were averaged separately to form a final score on the perceived social support from parents and perceived social support from friends. High scores indicate a greater perceived social support from parents and friends. Cronbach’s alphas ranged from 0.85 to 0.93.

### 2.4. Analytic Plan

A multivariate analysis of variance (MANOVA) was performed to examine age differences in cyberbullying involvement, as well as age and gender interactions, while controlling for face-to-face bullying involvement and technology use, for research question one. One multigroup comparison structural equation model with the Robust Maximum Likelihood estimator and the Full Information Maximum Likelihood approach to handle missing data were used to test research question two and three. Overall, roughly 0.5% of the data were missing, yielding 30 incomplete records, specifically 15 from the elementary school, five from the middle school, two from the high school, and eight from the university. Paths were added from cyberbullying involvement to perceived social support from friends and family and to depression. Gender was included as a predictor of cyberbullying involvement, perceived social support from friends and family, and depression, but it was not significant and dropped from further analyses. Two-way interactions were included between perceived social support from friends and cyberbullying involvement and between perceived social support from parents and cyberbullying involvement. Simple slopes were examined to determine the nature of the interaction. Technology use was controlled for in the analysis by allowing it to predict all forms of cyberbullying involvement; in addition, face-to-face bullying involvement was controlled for in the analysis by allowing it to predict all forms of cyberbullying involvement.

## 3. Results

### 3.1. Correlations

Correlations were performed among all of the study’s variables (see Table 1). Perceived social support from friends and family was positively associated with each other. In addition, perceived social support from friends and family were related negatively to cyberbullying victimization and cyberbullying perpetration among all of the participants. For middle school students only, perceived social support from friends and family were related negatively to witnessing cyberbullying. All forms of cyberbullying involvement were related to each other and to depression. Depression was negatively related to perceived social support from friends and family.

### 3.2. Differences in Cyberbullying Involvement

To answer research question one, a MANOVA was conducted (see Table 2 for means and standard deviations) to examine age group and gender differences in cyberbullying involvement, as well as interactions between age group and gender, while controlling for face-to-face bullying involvement and technology use. A main effect of the age group was found, Wilks’ Λ = 0.98, *F*(2,1293) = 479.63, *p* = 0.001, but gender as a main effect was not significant, Wilks’ Λ = 0.98, *F*(4,1299) = 0.96, *p* = 0.659. Findings revealed that middle school students were more often cyberbullies, cybervictims, and witnesses of cyberbullying, when compared to high school students, university students, and elementary school students. In addition, high school and university students did not differ for cyberbullying perpetration, victimization, and witnessing. High school and university students reported greater cyberbullying perpetration, victimization, and witnessing than elementary school students.

The interaction between age group and gender was significant, Wilks’ Λ = 0.98, *F*(6,1293) = 7.98, *p* = 0.036. To probe the interaction further, the analyses were split by age group and then conducted again, using a MANOVA. A main effect of gender was found for elementary school students and only for cyberbullying perpetration and victimization, but not for witnessing cyberbullying. Elementary school boys engaged in more cyberbullying perpetration and victimization, when compared to elementary school girls. For all other age groups, there were no significant findings regarding gender.

### 3.3. Association among Cyberbullying Involvement, Perceived Social Support, and Depression

To answer research questions two and three, the multigroup comparison structural model was performed and demonstrated an adequate fit, χ2(1601) = 591.76, *p* = 0.10, CFI = 0.98, TLI = 0.98, RMSEA = 0.04, SRMR = 0.03. Perceived social support from parents was associated negatively with depression for all age groups (see Table 3). In addition, perceived social support from friends was related negatively to depression for middle school and high school students. All types of cyberbullying involvement were related positively to depression among elementary school, middle school, high school, and university students.

Two-way interactions between cyberbullying involvement and social support from parents were significant across all age groups. Probing the interaction further revealed that cyberbullying involvement and depression were more strongly related at lower levels of perceived social support from parents, while opposite patterns were found for higher levels of perceived social support from parents. The two-way interactions between cyberbullying involvement and social support from friends were significant for middle and high school students, but not for elementary school and university students. The positive associations between cyberbullying involvement and depression were stronger for low levels of perceived social support from parents, but less positive for higher levels of perceived social support from parents.

## 4. Discussion

The purpose of this study was to examine age differences (i.e., elementary school, middle school, high school, and university students) in cyberbullying involvement, and whether perceived social support would moderate the associations between cyberbullying involvement and depression, as well as age differences in these relationships. This study contributes to a growing literature on the role of social support in mitigating the negative outcomes associated with cyberbullying perpetration, victimization, and witnessing.

### 4.1. Age Differences in Cyberbullying Involvement

Research question one involved examining age group and gender differences in cyberbullying involvement, including victimization, perpetration, and witnessing. Middle school students were more often involved in cyberbullying, when compared to other age groups. High school and university students did not differ in their cyberbullying involvement, while elementary school students had the lowest involvement. It is difficult to reconcile these findings with the literature on age group differences in cyberbullying involvement, due to mixed findings. In one of the only studies to examine age group differences in cyberbullying, Sevcikova and Smahel [17] found that early adolescents had the highest rates of cyberbullying perpetration and victimization when compared to younger and older age groups. Our finding that middle school students, who are early adolescents, had the highest rates of cyberbullying involvement is consistent with Sevcikova and Smahel’s [17] findings. However, other studies (e.g., ref. [18]) have found that other age groups were involved more often in cyberbullying. The conflicting findings between these other studies and the present study might be attributed to the measurement of cyberbullying, making it difficult to compare the studies. In addition, our study controlled for technology use and face-to-face bullying involvement, which is a methodological improvement over previous research on age group differences in cyberbullying involvement.

Gender interacted with age group as well, revealing that elementary school boys were more often cyberbullies and cybervictims, when compared to elementary school girls. The broader literature on gender differences in cyberbullying involvement is mixed, with some studies finding differences (e.g., refs. [8,21,22,23,24]) and others finding no gender differences [29,31,33]. Ultimately, we did not find any gender differences across most of the age groups. Oftentimes, technology use is a stronger indicator of being involved in cyberbullying, and it might be likely that regardless of age group, all adolescents and young adults are equally likely to experience, perpetrate, and witness cyberbullying [3,10,15]. For elementary school students, it might be likely that boys were more likely to interact with technology, increasing their risk of cyberbullying involvement.

### 4.2. Associations between Cyberbullying Involvement and Depression

Research question two involved investigating cyberbullying involvement in relation to depression, as well as age group differences. For all age groups, perceived social support from parents was negatively associated with depression and cyberbullying involvement (i.e., victimization, perpetration, witnessing) was related positively to depression. Such findings suggest that, regardless of age, believing one’s parents are there for them reduced participants’ vulnerability to cyberbullying involvement and depression [18,40,47]. Because parents are an active part of children’s, adolescents’, and young adults’ lives, it is highly probable that there are opportunities for parents to discuss ways to avoid online harm and provide recommendations on effective coping strategies for dealing with depression. Such a proposal is supported by research linking parental social support to lower levels of cyberbullying involvement and depression [23,34,36,48,54,55]. Parents provide advice and support, which helps reduce cyberbullying involvement and depressive symptoms [36,54]. Research suggests that parents provide strategies for ways to avoid online risks and that such discussions might reduce vulnerability to cyberbullying involvement and associated negative outcomes [36,55].

For middle school and high school students, perceived social support from friends was negatively associated with depression and cyberbullying involvement. Such findings highlight the incredible impact of supportive friendships on adolescents’ lives [18,40,47,56]. Friends, similar to parents, might provide opportunities to discuss strategies for avoiding online risks amongst each other. As with perceived social support from parents, supportive friends diminish adolescents’ cyberbullying involvement and depression through providing advice, strategies, and support for reducing risks and negative outcomes [23,36,54,55].

### 4.3. Buffering Effects of Perceived Social Support

Cyberbullying involvement is linked to depression, and other negative adjustment outcomes, including anxiety, suicidal ideation, non-suicidal self-harm, subjective health complaints, and substance use [8,25,57]. Cyberbullying involvement is a source of strain in children’s, adolescents’, and young adults’ lives, increasing their vulnerability to implementing negative coping strategies, including relying on revenge-focused coping strategies, to diminish or eliminate their negative feelings associated with cyberbullying [12]. Considering the likelihood that cyberbullying involvement is a source of strain and is linked to negative outcomes, it is important to investigate factors that might diminish such negative outcomes. Such a focus is important because it is unlikely that cyberbullying involvement can be completely avoided. Thus, a vital focus of this study was the buffering effect of perceived social support in the associations among cyberbullying involvement and depression, as well as age group differences (research question three). For all age groups, perceived social support from parents moderated the associations between cyberbullying involvement and depression, with higher levels of perceived social support reducing the positive relationship while lower levels of perceived social support increased the positive relationship.

The buffering effect of perceived social support from parents is supported by the literature on face-to-face bullying and cyberbullying involvement [58,59,60]. High levels of perceived social support from parents helps children, adolescents, and young adults realize that they have people in their lives, particularly parents, who care and support them [39,40,41]. When children, adolescents, and young adults perceived higher parental support, they are likely to feel efficacious when navigating negative situations, such as cyberbullying involvement. Such perceptions increase confidence in one’s abilities and potentially increases the likelihood of utilizing effective coping strategies [39,41]. Conversely, perceived low levels of social support from parents worsens negative outcomes following cyberbullying involvement. Children, adolescents, and young adults with lower perceived social support might not believe they have protection from negative situations, making them less secure when dealing with such situations. They might doubt their abilities to deal with negative situations effectively, like with cyberbullying involvement, and feel greater depression as a result.

Similar to the main effects of cyberbullying involvement and depression, perceived social support from friends moderated the relationships among cyberbullying involvement and depression, but for middle school and high school students only. Higher levels of perceived social support from friends diminished the positive relationship between cyberbullying involvement and depression, while lower levels of perceived social support had the opposite effect. Perceived social support from friends might function similarly as perceived social support from parents for adolescents. That is, friends, like parents, increase the likelihood that adolescents believe they have someone that is there for them through the tough times [36,54]. Relying on supportive friends is a type of social support that involves utilizing proactive coping strategies and perceiving high levels of social support from friends diminishes the positive association between cyberbullying involvement and depression, whereas lower social support reduces the likelihood of feeling supported and protected, increasing the susceptibility for negative outcomes, such as depression.

Although more similarities across age groups were found in cyberbullying involvement and for the buffering effect of perceived social support, a key finding from this study was the differential impact of parent and friend support on participants. In particular, for all participants, high perceived social support from parents reduced the positive relationship between cyberbullying involvement and depression, while lower levels increased this relationship. Such a pattern was found for perceived social support from friends as well, but only for middle school and high school students. It is difficult to reconcile this finding with the literature due to a lack of research on age differences in the buffering effects of perceived social support on the associations between cyberbullying involvement and depression. Research on age differences in social support might help to explain these findings. In this research, Jiang et al. [49] found that young adults (18–25 years) sought social support more than older adults (ages 60+) and Cao and colleagues [51] revealed that perceived social support was mediated by psychological capital during the pandemic for young adolescents (10–12 years) but not for young adults (18–25 years). In sum, the literature on age differences in perceived social support indicates that younger individuals versus older individuals rely more on social support. However, such a consensus does not indicate why we did not find buffering effects of perceived social support from friends for elementary school students. A potential explanation might be that such patterns and differences in perceived social support are found for adolescents and not other age groups. Follow-up research should investigate this proposal in more detail by utilizing a longitudinal design to identify changes overtime in perceived social support and its buffering effect on cyberbullying involvement and depression.

### 4.4. Limitations and Future Directions

Although a unique focus of the present study was to investigate age differences in cyberbullying involvement and the role of social support in the associations between cyberbullying involvement and depression, the study is cross-sectional, making it difficult to understand the longitudinal associations of the variables examined in this study. Follow-up research should include longer-term investigations and assess the variables examined in this study at multiple time points to understand the temporal ordering of perceived social support from parents and friends, cyberbullying involvement, and depression. It is also important for this research to examine changes over time in cyberbullying involvement, perceived social support, and depression by following the same age group over time. Furthermore, such follow-up research will help determine whether other variables, such as poverty and attachment to parents and peers, could have a role in the relationships examined in this study.

The questionnaire used to measure perceived social support from parents and friends provided a general assessment of social support that is more specific to the offline world. The participants in this study, particularly those of different age groups, might have held different opinions concerning the amount of social support they experience based on the social context. Furthermore, participants might perceive less social support in the online world versus the offline world. Future research should include an assessment of social support in the online world to better understand the buffering effect of perceived social support in the associations among cyberbullying involvement and depression.

The present study focused on depression as the only outcome of cyberbullying involvement. However, the literature provides evidence of various negative outcomes associated with cyberbullying involvement, including anxiety, suicidal ideation, non-suicidal self-harm, subjective health complaints, substance use, and academic problems. For example, researchers have found that academic performance varies based on non-bully/non-victim versus bully and victim designations [61]. Follow-up research should not only utilize longitudinal designs, but such studies should also examine other outcomes associated with cyberbullying involvement to examine age differences as well as how social support might buffer against other negative outcomes.

## 5. Conclusions

The present research is vital as it highlights the mechanisms that predict reductions in the negative effects of cyberbullying involvement across various age groups and identifies potential differences, while controlling for technology use and face-to-face bullying involvement. Results from this study revealed that middle school students were involved in cyberbullying more often than high school and university students, followed by elementary school students. High levels of perceived social support from parents diminished the positive relationships among cyberbullying involvement and depression, while low levels increased the relationship. Similar results were found for perceived social support from friends, but for middle school and high school students only. These findings have direct implications on programs designed to reduce cyberbullying involvement, and the potential of tailoring such programs based on age groups.

## Figures and Tables

**Table 1 ijerph-20-02835-t001:** Correlations among all variables.

	1	2	3	4	5	6
1. PSFriends	---					
2. PSFamily	0.44 *** 0.34 *** 0.31 *** 0.29 **	---				
3. CBV	−0.19 * −0.30 *** −0.25 ** −0.20 *	−0.25 ** −0.30 *** −0.20 * −0.20 *	---			
4. CBP	−0.22 * −0.38 *** −0.33 *** −0.20 *	−0.27 ** −0.33 *** −0.22 * −0.19 *	0.23 * 0.29 ** 0.27 ** 0.20 *	---		
5. CBW	−0.14 −0.18 * −0.13 −0.13	−0.13 −0.20 * −0.12 −0.09	0.25 ** 0.30 *** 0.26 ** 0.19 *	0.22 * 0.29 ** 0.25 ** 0.20 *	---	
6. Depression	−0.32 *** −0.32 *** −0.29 ** −0.31 ***	−0.33 *** −0.35 *** −0.30 *** −0.31 ***	0.30 *** 0.36 *** 0.34 *** 0.30 ***	0.30 *** 0.33 *** 0.28 ** 0.28 **	0.28 ** 0.30 *** 0.29 ** 0.26 *	---

PSFriends = perceived social support from friends; PSFamily = perceived social support from family; CBV = cyberbullying victimization; CBP = cyberbullying perpetration; CBW = cyberbullying witnessing. First number corresponds to elementary school students; second number corresponds to middle school students; third number corresponds to high school students; fourth number corresponds to university students. * *p* < 0.05. ** *p* < 0.01. *** *p* < 0.001.

**Table 2 ijerph-20-02835-t002:** Means and standard deviations for cyberbullying involvement by age group.

	Elementary School	Middle School	High School	University
	*M* (*SD*)	*M* (*SD*)	*M* (*SD*)	*M* (*SD*)
CBV				
Overall	2.59 (0.61)	3.74 (0.65)	2.91 (0.65)	2.93 (0.61)
Males	2.68 (0.69)	3.77 (0.60)	2.88 (0.69)	2.95 (0.60)
Females	2.49 (0.52)	3.71 (0.70)	2.93 (0.60)	2.91 (0.62)
CBP				
Overall	2.76 (0.61)	3.28 (0.55)	3.11 (0.60)	3.11 (0.65)
Males	2.90 (0.70)	3.33 (0.60)	3.12 (0.59)	3.13 (0.69)
Females	2.61 (0.51)	3.22 (0.50)	3.09 (0.61)	3.09 (0.61)
CBW				
Overall	2.89 (0.66)	3.63 (0.82)	3.45 (0.75)	3.43 (0.73)
Males	2.91 (0.71)	3.65 (0.83)	3.40 (0.71)	3.27 (0.65)
Females	2.86 (0.61)	3.60 (0.80)	3.49 (0.79)	3.59 (0.80)

CBV = cyberbullying victimization; CBP = cyberbullying perpetration; CBW = cyberbullying.

**Table 3 ijerph-20-02835-t003:** Multigroup comparison of the associations among perceived social support, cyberbullying involvement, and depression by age group.

		Depression	
Country	Predictors	β	*SE*
Elementary School	PSFriends	−0.16	0.04
	PSFamily	−0.18 *	0.05
	CBV	0.29 ***	0.10
	CBP	0.24 **	0.09
	CBW	0.22 *	0.07
	PSFriends × CBV	0.01	0.01
	PSFriends × CBP	0.02	0.01
	PSFriends × CBW	0.02	0.01
	PSFamily × CBV	0.16 **	0.04
	PSFamily × CBP	0.16 **	0.05
	PSFamily × CBW	0.18 **	0.05
Middle School	PSFriends	−0.28 **	0.10
	PSFamily	−0.25 **	0.09
	CBV	0.33 ***	0.12
	CBP	0.30 ***	0.10
	CBW	0.33 ***	0.11
	PSFriends × CBV	0.22 **	0.08
	PSFriends × CBP	0.20 **	0.06
	PSFriends × CBW	0.20 **	0.06
	PSFamily × CBV	0.19 *	0.07
	PSFamily × CBP	0.18 *	0.06
	PSFamily × CBW	0.19 *	0.06
High School	PSFriends	−0.26 **	0.09
	PSFamily	−0.21 *	0.06
	CBV	0.30 ***	0.11
	CBP	0.27 **	0.10
	CBW	0.31 ***	0.12
	PSFriends × CBV	0.21 **	0.07
	PSFriends × CBP	0.19 *	0.06
	PSFriends × CBW	0.19 *	0.07
	PSFamily × CBV	0.20 **	0.07
	PSFamily × CBP	0.17 *	0.06
	PSFamily × CBW	0.18 *	0.06
University	PSFriends	−0.14	0.04
	PSFamily	−0.20 *	0.08
	CBV	0.29 ***	0.10
	CBP	0.26 **	0.09
	CBW	0.20 *	0.08
	PSFriends × CBV	0.03	0.01
	PSFriends × CBP	0.04	0.02
	PSFriends × CBW	0.03	0.01
	PSFamily × CBV	0.17 **	0.07
	PSFamily × CBP	0.16 **	0.06
	PSFamily x CBW	0.20 **	0.08

PSFriends = perceived social support from friends; PSFamily = perceived social support from family; CBV = cyberbullying victimization; CBP = cyberbullying perpetration; CBW = cyberbullying witnessing. * *p* < 0.05. ** *p* < 0.01. *** *p* < 0.001.

## Data Availability

Data can be requested from the first author.

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
