# Peer review of "Cyberbullying Involvement and Depression among Elementary School, Middle School, High School, and University Students: The Role of Social Support and Gender"

_ijerph, 2023, doi:10.3390/ijerph20042835_

Round 1

Reviewer 1 Report

This is an interesting and important article, with a clear empirical focus and methodology.

It might be the literature, but when you're talking about 'gender' it seems to me you're talking about (biological) sex, i.e. male or female. As I understand it, 'gender' is the term for the more psychological and cultural aspects, so girls and boys, women and men, masculine and feminine, the concept man and woman (e.g. de Beauvoir). 'Sex' is reserved for the more biological, of which there are two dominant ones, i.e. male and female. It seems in this literature and article 'gender' is used for the both? If so I guess it's fine, but I would maybe add a note so as not to confuse readers from other disciplinary backgrounds (feminism, social studies, philosophy etc.).

Author Response

Comment: This is an interesting and important article, with a clear empirical focus and methodology.

Response: Thank you! We appreciate your review.

Comment: It might be the literature, but when you're talking about 'gender' it seems to me you're talking about (biological) sex, i.e. male or female. As I understand it, 'gender' is the term for the more psychological and cultural aspects, so girls and boys, women and men, masculine and feminine, the concept man and woman (e.g. de Beauvoir). 'Sex' is reserved for the more biological, of which there are two dominant ones, i.e. male and female. It seems in this literature and article 'gender' is used for the both? If so I guess it's fine, but I would maybe add a note so as not to confuse readers from other disciplinary backgrounds (feminism, social studies, philosophy etc.).

Response: We specifically asked adolescents “What is your gender?” We did not specifically include this in the methods, unfortunately, although we did indicate that we asked participants demographic information, including gender. We have revised the last paragraph in the Procedures section to clarify that we specifically asked, “What is your gender?” and then participants responded with their answers. We did not have any classification other than male/female, boy/girl, or man/woman. We hope that it clarifies our conceptualization of this variable. The studies we reviewed on gender specify boy/girl designations and also use “gender” as terminology. We hope this is satisfactory.

Reviewer 2 Report

  The study, although it is interesting, needs many modifications to be a scientific article. Regarding the introduction, add this quote: https://doi.org/10.3390/ijerph19159301 The instructions for permission to sleep are too extensive. Resume. As for the methodology, it is very complex, and I think it should be divided into several articles, or something similar. There are questionnaires that are not validated, although Cronbach's alpha is indicated, so the authors should consider validating these questionnaires if they want to use them, reliability and validity are not the same. The statistics are carried out in a complex way, and the presentation is confusing, since it is not understood to do a manova, if the value of p is not used, and the post hoc, only with descriptive data. Table 1 is very confusing, since the explanation is not very clear. Table 4 is not understood at all. For all these reasons, it would be interesting to validate the scales in the first place, to know what the values of each created scale are, and to know what a score means, and if it is high or low, and then make an appropriate statistic with the data. Thank you.

Author Response

Comment: The study, although it is interesting, needs many modifications to be a scientific article.

Response: Thank you for the comments. We appreciate the comments to improve the manuscript.

Comment: Regarding the introduction, add this quote: https://doi.org/10.3390/ijerph19159301

Response: Thank you for the suggestion. We think this article fits well in our future directions section when we talk about examining academic performance. We have added it to this section and added the reference to the reference list.

Comment: The instructions for permission to sleep are too extensive. Resume.

Response: We hope we understand this comment. It is important to be thorough about parental permission slips provided to minors to maintain APA standards. However, we do agree that the specific totals of permissions, which is included in the second paragraph of the Methods, in the fourth sentence, can be removed. If someone had questions about the total of permission slips, they could easily add up the amounts in the next sentence. We hope this answers your concern.

Comment:  As for the methodology, it is very complex, and I think it should be divided into several articles, or something similar.

Response: It is complex because we are assessing multiple variables across various age groups. Thus, there is a lot to describe regarding samples and measures. We did go through the methodology section to remove information that might not necessarily be important for the manuscript.

Comment: There are questionnaires that are not validated, although Cronbach's alpha is indicated, so the authors should consider validating these questionnaires if they want to use them, reliability and validity are not the same.

Response: The Cronbach’s alphas are similar to those in the studies using the same measures. In addition, the correlations are also in the expected direction and are similar in terms of the relationships found in other studies as well. The questionnaires are validated.

Comment: The statistics are carried out in a complex way, and the presentation is confusing, since it is not understood to do a manova, if the value of p is not used, and the post hoc, only with descriptive data. Table 1 is very confusing, since the explanation is not very clear. Table 4 is not understood at all. For all these reasons, it would be interesting to validate the scales in the first place, to know what the values of each created scale are, and to know what a score means, and if it is high or low, and then make an appropriate statistic with the data. Thank you.

Response: We do not have a Table 4 so are not certain which table is in question here regarding one part of your comment. Table 1 presents the correlation matrix. It is complex, but please remember that we have four separate samples of different age groups. We believe maybe adding some borders will clarify what the correlations correspond to. We hope this helps with the clarify of this table.